# Application of Spectroscopic Methods for the Identification of Superoxide Dismutases in Cyanobacteria

**DOI:** 10.3390/ijms232213819

**Published:** 2022-11-10

**Authors:** Monika Kula-Maximenko, Kamil Jan Zieliński, Joanna Depciuch, Janusz Lekki, Marcin Niemiec, Ireneusz Ślesak

**Affiliations:** 1The Franciszek Górski Institute of Plant Physiology, Polish Academy of Sciences, Niezapominajek 21, 30-239 Krakow, Poland; 2Institute of Nuclear Physics, Polish Academy of Sciences, 31-342 Krakow, Poland; 3Department of Agricultural and Environmental Chemistry, University of Agriculture in Krakow, 31-120 Krakow, Poland

**Keywords:** AES, FTIR, metalloproteins, oxygenic photosynthesis, phylogeny, PIXE

## Abstract

Superoxide dismutases (SODs) belong to the group of metalloenzymes that remove superoxide anion radicals and they have been identified in three domains of life: *Bacteria*, *Archaea* and *Eucarya*. SODs in *Synechocystis* sp. PCC 6803, *Gloeobacter violaceus* CCALA 979, and *Geitlerinema* sp. ZHR1A were investigated. We hypothesized that iron (FeSOD) and/or manganese (MnSOD) dominate as active forms in these cyanobacteria. Activity staining and three different spectroscopic methods of SOD activity bands excised from the gels were used to identify a suitable metal in the separated samples. FeSODs or enzymes belonging to the Fe-MnSOD superfamily were detected. The spectroscopic analyses showed that only Fe is present in the SOD activity bands. We found FeSOD in *Synechocystis* sp. PCC 6803 while two forms in *G. violaceus* and *Geitlerinema* sp. ZHR1A: FeSOD1 and FeSOD2 were present. However, no active Cu/ZnSODs were identified in *G. violaceus* and *Geitlerinema* sp. ZHR1A. We have shown that selected spectroscopic techniques can be complementary to the commonly used method of staining for SOD activity in a gel. Furthermore, the occurrence of active SODs in the cyanobacteria studied is also discussed in the context of SOD evolution in oxyphotrophs.

## 1. Introduction

The superoxide anion radical (O_2_^•_−_^) is one of the major reactive oxygen species (ROS) produced in living cells as a result of the one-electron reduction of molecular oxygen (O_2_):O_2_ + e^−^ → O_2_^•_−_^
(1)

The superoxide anion radical is spontaneously generated mainly in some oxidase-dependent reactions and cellular charge-transfer processes, including the photosynthetic and respiratory electron transport chain [1,2,3,4]. O_2_^•_−_^ itself is not very toxic for cellular organic molecules, but it can be a source of other, more toxic ROS, especially hydroxyl radicals (HO^•^), which are very harmful to most biomolecules [1,2]. To protect cells from the toxic effects of HO^•^, superoxide dismutases (SODs, EC 1.15.1.1) scavenge O_2_^•_−_^ according to the summarized equation [5,6,7,8,9]:2O_2_^•_−_^ + 2H^+^ → H_2_O_2_ + O_2_
(2)

SODs are found in all three domains of life: *Bacteria*, *Archaea* and *Eucarya* and are metalloproteins that are among the key enzymes involved in the antioxidant response system [7]. More broadly, SODs, along with other antioxidant enzymes, play a crucial role in redox regulation and maintenance of cellular homeostasis [10,11]. In general, based on a metal ion in the active site, four different classes of SOD have been identified: manganese SOD (MnSOD), iron SOD (FeSOD), copper/zinc SOD (Cu/ZnSOD) and nickel SOD (NiSOD). An exception to this typology are the SODs of some bacterial species that are active in the presence of both metals, i.e., Mn and Fe, and are referred to as cambialistic SODs [6,7,12,13]. In contrast, Co/Zn- and Fe/ZnSODs have been identified in bovine liver and the bacterium *Streptomyces coelicolor*, respectively [14,15]. In eukaryotic cells, the different forms of SOD are located in separate intracellular compartments. FeSOD can be found in chloroplasts and peroxisomes; MnSOD is mainly located in mitochondria and peroxisomes, while Cu/ZnSOD has been detected in chloroplasts, mitochondria, nucleus and extracellular matrix [6,7,13,16].

Although the activity of SODs is closely linked to O_2_-utilizing metabolism, their presence, at least at the genomic level, has been confirmed even in the majority of organisms classified as obligate anaerobes [9,17,18,19]. Therefore, phylogenetic analyses have suggested that SODs may have appeared in the very early stages of the evolution of life on Earth, long before the great oxidation event (GOE) about 2.4 Gyr ago [9,19,20]. To the best of our knowledge, cyanobacteria or cyanobacteria-like organisms were the first entities capable of oxygenic photosynthesis on Earth [21,22,23,24]. Photosynthetic O_2_ production was inextricably linked to ROS production, including O_2_^•_−_^ [3,25,26]. Therefore, the identification and characterization of SODs in cyanobacteria also gives us insight into the evolution of O_2_-utilizing metabolic pathways, including oxygenic photosynthesis.

In the studies presented here, the identification of active SOD forms in *Synechocystis* sp. PCC 6803 (hereafter referred to as *Synechocystis*), *Gloeobacter violaceus* CCALA 979 (hereafter referred to as *G. violaceus*), *Geitlerinema* sp. ZHR1A was performed by comparing experimental methods for identification SOD and in silico analyzes based on SOD sequencing data publicly available in biological databases. *Synechocystis* is a model microorganism used in studies dealing with the photosynthetic and respiratory electron transport chain in cyanobacteria [27]; *G. violaceus* is considered to be the most primordial cyanobacterium without thylakoid membranes and with a photosynthetic apparatus localized in the cytoplasmic membrane [28,29,30,31], while *Geitlerinema* sp. ZHR1A is a filamentous cyanobacterium compared to the previously mentioned species, whose genome has not yet been sequenced and whose physiology is poorly known.

The formulated hypothesis was that Fe and/or MnSODs dominate as active forms in the cyanobacteria studied. However, in addition to FeSOD/MnSOD, Cu/ZnSODs are also present in some cyanobacterial genomes. We have used selected spectroscopic techniques to identify forms of SOD that complement the method of staining SOD activity in a gel. The results showed that FeSOD and/or their isoforms are active in *Synechocystis*, *G. violaceus*, and *Geitlerinema* sp. ZHR1, while active Cu/ZnSODs were not identified at all. We also discuss the data based on the evolution of SODs.

## 2. Results

### 2.1. The 16S rRNA Tree and the Relationship between the Cyanobacteria Studied

A phylogenetic tree was reconstructed for a small sample of selected 16S rRNA from different organisms that perform oxygenic photosynthesis. Reconstruction of the 16S rRNA tree showed that strain *G. violaceus* CCALA 979 is closely related to *G. violaceus* PCC 7421, whose genome was sequenced (Figure 1). This result also confirms the basal phylogenetic position of *G. violaceus* CCALA 979 in the Gloeobacter clade together with *G. kilaueensis* [32]. This does not mean that the genome of *G. violaceus* CCALA 979 is identical to that of *G. violaceus* PCC 7421, but it does suggest that these genomes are very similar, which could indicate the possible presence of the same genes encoding the different forms of SOD. The 16S rRNA sequence of *Geitlerinema* ZHR1A was in a group with *Synechocystis* sp. PCC 6803, which was studied in this work, and *Microcystis aeruginosa* PCC 9701 (Figure 1).

### 2.2. Identification of SODs

Based on sequences of SODs available in the UniProt database (www.uniprot.org, 3 November 2021), the presence of FeSOD was experimentally confirmed only in *Synechocystis*. In the case of *G. violaceus* and *Geitlerinema* sp. ZHR1A, we cannot identify which form SOD is present in these cyanobacteria if we use standard staining without inhibitors for selected forms of SOD. Therefore, to test this, activity staining of SOD was performed with inhibitors, i.e., H_2_O_2_ and KCN in a gel (Figure 2). In general, Cu/ZnSOD is inhibited by H_2_O_2_ and cyanide, while FeSOD is inhibited only by H_2_O_2_. MnSOD is inhibited neither by H_2_O_2_ nor by CN^−^ [7].

A single active form of FeSOD (band 1A) was identified in *Synechocystis* (Figure 2). The estimated molecular weight of this form was approximately 40 kDa (Figure 2, Appendix A). The result obtained supports the data from literature showing the in-gel activity staining and protein content for FeSOD (*sodB*) in *Synechocystis* [33,34].

The inhibitor assay showed that two distinct SOD activity bands were identified in *G. violaceus*, designated 1B and 2B. SOD band 1B with a molecular weight of about 43 kDa was much more intense than SOD band 2B with a molecular weight of about 63 kDa (Figure 2, Appendix A). We observed the incomplete inhibition of SOD activity of band 1B by H_2_O_2_ (Figure 2a). This could indicate that: (i) it is the most likely FeSOD isoform (1B) but not completely inhibited by H_2_O_2_, (ii) it is MnSOD that is partially sensitive to H_2_O_2_, or (iii) the band represents a form of cambialistic Fe/MnSOD. Therefore, band 1B was designated as FeSOD/MnSOD after the activity staining experiment (Figure 2). Band 2B was strongly inhibited by H_2_O_2_, and it was identified as FeSOD (Figure 2).

In the case of *Geitlerinema* sp. ZHR1A, two SOD bands were detected and designated as 1C and 2C. Based on activity staining with inhibitors, bands 1C (ca. 40 kDa) and 2C (ca. 85 kDa) were identified as FeSOD and MnSOD, respectively (Figure 2, Appendix A).

It should also be emphasized that the test with inhibitors did not reveal any Cu/ZnSOD form in the cyanobacteria examined (Figure 2a). In order to exclude or confirm the presence of a specific metal in the identified SOD activity bands, especially in cases where the inhibitor assay does not give clear results, we applied selected spectroscopic methods.

### 2.3. Metal Identification by the Spectroscopic Methods

Spectroscopic methods were used to identify the metals present in the excised gel sections: 1A, 1B, 1C, 2B, and 2C that corresponded to the forms of SOD, which were identified in a test with inhibitors as: MnSOD and FeSOD (Figure 2).

#### 2.3.1. Fourier Transformed Infrared (FTIR) Spectroscopy

The Fourier transformed infrared spectroscopy FTIR spectra of *Synechocystis*, *G. violaceus* and *Geitlerinema* sp. ZHR1A are shown in Figure 3a–c, respectively. In each of the FTIR spectra shown in Figure 3, the absorption bands at about 1400–1500 cm^−1^ are visible, which are attributed to asymmetric bending vibrations of the -CH_2_, -CH_3_ groups [35]. Moreover, the peaks at 1650 and 1545 cm^−1^ correspond to amide I (C=O stretching mode) and amide II (combination of N-H bending and C-N stretching mode) of SOD [36] and the peaks in the range 1681–1810 cm^−1^ and at the wavenumber 1361 cm^−1^ were attributed to the carbonyl (C=O) and methyl (C-H of -CH_3_) groups of synthetic iron-mineralized SOD (SOD-Fe^0^@Lapa-Z) [37]. In general, these peaks are the most sensitive to changes in the structure and chemical composition of MnSOD [38]. However, for MnSOD, the most typical band is from the amide I at 1657 cm^−1^ with the shoulder at 1627 cm^−1^, which is specific to the helical protein structure [39]. In our tested SOD samples, the MnSOD band was not found at its characteristic wavelength (1657 cm^−1^), and even a drop of the spectrum was visible in this band (Figure 3). The amide II band has components at 1515 cm^−1^, 1530 cm^−1^ and 1545 cm^−1^, which are in overlapping regions of tyrosine side-chain, random, β-sheet and α-helical motives, respectively [40]. Moreover, the IR amide III band (1200–1340 cm^−1^) is also visible [40,41]. In addition, the bands characteristic of CuZnSOD at 1622 cm^−1^, 1631 cm^−1^, 1669 cm^−1^ and 1679 cm^−1^ were not found [42]. The characteristic peak of NiSOD was also not detected in the FTIR spectrum [43].

FTIR spectroscopy did not clearly show which metal was present in the samples studied, as the available literature data concern FTIR spectra for MnSOD [38,39]. However, MnSOD is structurally closely related to FeSOD and both belong to the same superfamily of proteins [7]. In our opinion, the FTIR data in combination with the SOD activity staining (Figure 2) indicate that the metals in the analyzed bands of the gel are most likely iron or manganese.

In a next step, we decided to use the more precise spectroscopic methods, i.e., particle-induced X-ray emission (PIXE) and atomic emission spectroscopy (AES), to identify the metal that was present in the gel bands with SOD activity. To provide more clarity, the following designations were given to the SOD forms of each gel band: FeSOD1 for 1A, 1B, and 1C; FeSOD2 for 2B. We have designated band 2C, previously referred to as MnSOD based on the activity staining (Figure 2), as FeSOD2 (Figure 3), as further spectroscopic analysis excluded Mn ions in this band. Therefore, the same notation was used later in the paper for the PIXE and AES (see below).

#### 2.3.2. Particle-Induced X-Ray Emission (PIXE) and Atomic Emission Spectroscopy (AES)

The techniques PIXE and AES were used to determine the microscopic quantities of elements. The sensitivity of these techniques (<µg g^−1^) makes it possible to measure trace values of the elements. Qualitative analysis is a simple task for PIXE because the energy of the emitted X-rays depends on the atomic number of elements according to Moseley’s law. Peaks in the spectrum occur at the energies of the characteristic X-rays of the individual elements. We measured the spectra of the individual activity bands of SOD in the gel and the spectrum of the unstained gel to determine the elemental composition of the background (see “Materials and Methods”). K and Ca were identified in the pure gel. In the investigated bands of SOD, the PIXE analysis identified only Fe (Kα—6.40 keV), while the presence of the Mn (Kα—5.90 keV) was not detected (Figure 4).

Among the tested SOD’s bands, the first form of SOD (FeSOD1) had the highest amount of Fe in the tested strains (Table 1). In *Geitlerinema* sp. ZHR1A and *G. violaceus*, the FeSOD1 band contained approximately 10 and 8 µg g^−1^ Fe, respectively, while the band of *Synechocystis* sp. PCC 6803 contained only 5 µg g^−1^. The second band (FeSOD2) was detected in *Geitlerinema* sp. ZHR1A and in *G. violaceus* and contained a lower amount of Fe: approx. 3 µg g^−1^ in *Geitlerinema* sp. ZHR1A and 1 µg g^−1^ in *G. violaceus*. Mn was not found in any of the gel bands examined (Table 1).

PIXE and AES methods showed that the activity bands of SOD were enriched in Fe and no other metals specific to SOD, including Mn, Cu, Zn, and Ni, were detected in the analyzed samples of all cyanobacterial species (Figure 3 and Figure 4, Table 1).

## 3. Discussion

Cyanobacteria-like organisms, as the first O_2_ producers, most likely had to be equipped with SODs [20,44]. Genes encoding different classes of SOD are present in contemporary cyanobacteria, and the relevant data are collected in publicly available biological databases. However, most proteins are classified as SODs based on orthologs that exist in closely related species or are predicted based only on automatic annotation. A minority of cyanobacterial SODs have been detected at the experimental level, e.g., by detection of appropriate transcripts and/or proteins (www.uniprot.org, 3 November 2021).

We have confirmed the presence of a single form of FeSOD (about 40 kDa) in *Synechocystis* by all analytical methods used here, which is consistent with both literature data and databases [33,34]. This shows that the detection methods used to identify SOD were adequate and provide reliable information. Therefore, FeSOD1 from *Synechocystis* can be considered as the “reference FeSOD”. In the majority of cyanobacteria: *Spirulina platensis*, *Microcystis aeruginosa*, *Nostoc* PCC 7120, *Nostoc commune*, *Oscillatoria* sp. the occurrence of Fe- or/and MnSOD was identified [45,46,47,48,49]. The failure of H_2_O_2_ to completely inhibit FeSOD2 in *G. violaceus* and particularly in *Geitlerinema* sp. ZHR1A may also indicate the existence of MnSOD. Although spectroscopy studies excluded Mn ions in the samples, it should be emphasized that in *Propionibacterium shermanii* and *Methanobacterium thermoautitrophicum* Fe-containing SODs were not sensitive to inhibitory effects of H_2_O_2_ [50,51].

The complete lack of detectable activity of Cu/ZnSODs in *G. violaceus* and *Geitlerinema* sp. ZHR1A was noted. One possible answer is that these Cu/ZnSODs represent inactive genes/proteins for some reason or a detection method, e.g., in a gel activity stain, is not sufficiently accurate and sensitive. However, we did not find any data in the literature that concern pseudogenes of SODs or a more accurate method for SOD identification in crude cell extracts of any organisms. Nevertheless, it cannot be excluded that sequences predicted to belong to the Cu/ZnSOD family are pseudogenes. Pseudogenes are present in the genome of cyanobacteria and their levels vary among species and strains. Many genes encoding antioxidant enzymes, such as catalase, glutathione peroxidase, and glutathione S-transferase, have been classified as pseudogenes in cyanobacteria [52]. In the case of *G. violaceus* CCALA 979 and *Geitlerinema* sp. ZHR1A, their genomes have not been sequenced. This result is particularly interesting for *G. violaceus* CCALA 979, as it is closely related to *G. violaceus* PCC 7421 and *G. kilaueensis* (Figure 1) [32,53]. According to the UniProt database (www.uniprot.org, 3 November 2021), both *G. violaceus* PCC 7421 and *G. kilaueensis* have genes for Cu/ZnSOD-like proteins. It is possible that the genes encoding Cu/ZnSOD (*sodC*) were not transcribed and translated into active CuZnSOD under the culture conditions tested. Several studies have shown that SOD genes are expressed in response to changing environmental conditions [54,55].

It has been shown that Cu/ZnSODs are rather rare SODs in cyanobacteria compared to Fe- and MnSODs. Nevertheless, a possible involvement of Cu/ZnSODs as an evolutionarily older class of SOD in cyanobacteria than FeSODs has recently been postulated [20]. Our data support previous findings that FeSODs are at least as old as Cu/ZnSODs or even older than Cu/ZnSODs among SODs [7,8,44,56]. Especially due to the basal position of the 16S rRNA of *G. violaceus* CCALA 979, together with other cyanobacteria from Gloeobacter clade (Figure 1) and its active FeSODs, an ancient origin of this class of SOD in cyanobacteria cannot be excluded. Moreover, the presence of iron in many other antioxidant enzymes in *Archaea*, *Bacteria*, and *Eucarya* may suggest that these enzymes evolved before GOE and the release of significant amounts of O_2_ into the environment [17,19]. In our opinion, the possible antiquity of Cu/ZnSOD in ancient cyanobacteria, such as *Gloeobacter*, needs further studies.

The obtained results of SOD activity staining, supported by spectroscopic analyses, provided evidence that the only active class of SODs in *Synechocystis* sp. PCC 6803, *G. violaceus* CCALA 979 and *Geitlerinema* sp. ZHR1A is FeSOD.

## 4. Materials and Methods

### 4.1. Cyanobacteria

Three species of cyanobacteria were analyzed: *Synechocystis* sp. PCC 6803, *G. violaceus* and *Geitlerinema* sp. ZHR1A, which differ in their morphology and ecological habitats. The strain *Synechocystis* sp. PCC 6803 was acquired from Collections des Cyanobactéries of the Institut Pasteur in Paris (France). The other two strains *G. violaceus* (CCALA 979) and *Geitlerinema* sp. ZHR1A (CCALA 1069) were acquired from the Culture Collection of Autotrophic Organisms (CCALA) of the Institute of Botany CAS in Třeboň (Czech Republic). For *G. violaceus* CCALA 979, the reference strain is *G. violaceus* PCC 7421 [53], which was first isolated as *G. violaceus* strain 7421 by Rippka et al. [57].

### 4.2. Databases Query and Phylogenetic Analysis

In contrast to *Synechocystis*, the genomes of *G. violaceus* CCALA 979 and *Geitlerinema* sp. ZHR1A have not been sequenced. Therefore, the relationship between the strains studied and the species whose genomes have been sequenced was analyzed using the sequenced 16S rRNA fragments of *G. violaceus* and *Geitlerinema* sp. ZHR1A as well as 16S rRNA sequences from databases.

#### 4.2.1. Sequencing of 16S rRNA

Total RNA was isolated from 0.5 g of lyophilized cyanobacteria samples using the RNeasy Plant Mini Kit (QIAGEN) according to the manufacturer’s instructions. The RNA was then transcribed into cDNA; cDNA was obtained from 5 µg RNA using the TranScriba Kit (A&A Biotechnology), according to the manufacturer’s instructions. Universal primers for amplification of 16S rRNA in cyanobacteria were used to find the 16S rRNA fragments [58]; forward primer CYA 359F: 5′-GGGGAATYTTCCGCAATGGG-3′and reverse primer CYA 781R: 5′-GACTACWGGGGTATCTAATCCCWTT-3′. The PCR products obtained were subjected to sequencing (Genomed S.A., Warszawa, Poland). The 16S rRNA sequences (fragments) are shown in Appendix A. Subsequently, the sequences were used for the reconstruction of the 16S rRNA tree.

#### 4.2.2. The Reconstruction of 16S rRNA Tree

All 16S rRNA sequences were derived from Silva (http://www.arb-silva.de, 3 November 2021) and GenBank (www.ncbi.nlm.nih.gov, 3 November 2021) (for details see Appendix A). The sequences were aligned with the default parameters of MUSCLE implemented in MEGA6 [59]. The evolutionary history was inferred by using the Maximum Likelihood (ML) method based on the Kimura 2-parameter model [60]. A discrete gamma distribution was used to model evolutionary rate differences among sites. The bootstrap consensus tree was inferred from 500 replicates [61]. The tree is drawn to scale, with branch lengths measured in the number of substitutions per site. This analysis involved 24 nucleotide sequences. The phylogenetic reconstruction was performed in MEGA6 [59]. 

### 4.3. Growth Conditions

The strains were cultured in BG11 medium [62] in 500 mL Duran bottles. The bottles were flushed with air using an aquarium membrane pump (SERA Air 550 R Plus, Heinsberg, Germany). For culture, 16 mL of the cyanobacterial suspension was added to the 400 mL of medium. The cultures of *Synechocystis* sp. PCC 6803 and *Geitlerinema* sp. ZHR1A were grown for 15 days, while the culture of *G. violaceus* was cultured for 30 days at a photoperiod of 16h:8h, light:dark at 25 ± 1 °C under white light from LED lamps. The LED’s light intensity was established at 70 μmol photons m^−2^ s^−1^. On these days, the cyanobacteria culture had a maximum increase in biomass, the measure of which was the OD (optical density) at 730 nm (OD_730_) between about 1.0 and 2.0. Samples for SOD analyses were taken between the 4th and 5th hour of the photoperiod.

### 4.4. Preparation of Soluble Proteins from Cyanobacteria

In total, 15 mL of the culture was centrifuged at 1800 g for 10 min and the pellet was lyophilized. The resulting pellet was then frozen in liquid nitrogen and thawed in a sonic bath, allowing the cells to partially break down. This was done three times. Metal beads were added to the pellet and triturated at a frequency of 25 s^−1^ for 3 min. Then, 500 μL of homogenization buffer consisting of 100 mM Tricine-Tris, 3 mM MgSO_4_ 7H_2_O, 1 mM DTT, 3 mM EGTA, pH 8.0, and cOmplete™ Protease Inhibitor Cocktail tablets (Roche) were added to ca. 50 mg of the biomass powder. The suspension was again triturated with metal beads and centrifuged at 22,000 g for 10 min and the supernatant fraction was collected. Protein concentration was determined according to Bradford [63] using the Bio-Rad Protein Assay according to the manufacturer’s instructions. The protein extract was loaded directly onto a native polyacrylamide gel.

### 4.5. SOD Assay

The SOD forms were separated by native electrophoresis on 12% (*w*:*v*) polyacrylamide gels at a constant voltage of 180 V for about 1 h [64]. A total of 30 µg of protein was loaded onto each lane. The gels were stained for SOD activity according to the method described by Beauchamp and Fridovich [65]. After electrophoresis and incubation for 20–25 min in the dark at room temperature in the standard staining buffer: 50 mM phosphate buffer, pH 7.8, 10 mM EDTA, 28 mM TEMED, 30 µM riboflavin and 245 µM nitro blue tetrazolium chloride (NBT), the gels were exposed to the artificial white light until the SOD activity bands became visible. To inhibit Cu/ZnSOD and FeSOD, the gels were stained in a standard buffer containing 5 mM H_2_O_2_. Inhibition of Cu/ZnSOD was achieved in the presence of 2 mM KCN. In addition, a protein mass ladder (PageRuler™ Prestained Protein Ladder, Thermo Scientific, Waltham, MA, USA) was used to estimate the molecular weight (MW) of the separated SOD forms (see also Appendix A).

### 4.6. Quantitative Analysis

Images of SOD staining on native gels were acquired by photographic documentation using the GeneSnap system (Cambridge, UK) and densitometric analysis of bands was performed using ImageJ software (NIH, Bethesda, MD, USA) for semi-quantitative determination of SOD activity.

### 4.7. Metal Identification in SOD Activity Bands by Spectroscopic Methods

In order to additionally verify which type of metal is present in the identified SODs, the SOD activity bands were cut out of a gel (Figure 5) and subsequently analyzed using various spectroscopic methods described below.

### 4.8. Fourier Transformed Infrared (FTIR) Spectroscopy

For spectroscopic measurements, Vertex 70v from Bruker was used. The samples detection was made with liquid-nitrogen cooled mercury cadmium telluride (MCT) and diamond attenuated total reflection (ATR) crystal plate. IR spectral resolution were acquired at 2 cm^−1^ with 32 scans within the ranges 4000 cm^−1^ to 400 cm^−1^. Prior to each sample measurement, the empty diamond/MCT of the ATR unit was recorded as background and repeatedly subtracted via the appropriate software to eliminate water in the sample and air. To measure the FTIR spectrum, the sample was placed directly on the diamond crystal of the ATR device. The crystal of the instrument was cleaned and dried before and after each sample’s measurements. After waiting for approximately 5 min (the water evaporated from the sample), the measurement was taken. The FTIR spectra were measured for five samples. For each sample, three spectra were recorded. Thus, 15 spectra were obtained. Then, the three spectra were averaged for each sample. After pre-processing, the selected spectral band regions were analyzed using the OPUS setup to detect the changes in the samples. For all spectra, the baseline correction vector was normalized, 7 points were smoothed with Savitzky–Golay and ATR correction was applied.

### 4.9. Particle-Induced X-ray Emission (PIXE)

The trace elements in the investigated samples were analyzed by detecting of characteristic X-ray radiation (PIXE—particle-induced X-ray emission technique) at the proton microprobe [66] at the Van de Graaff accelerator.

For the elemental analysis, the individual SOD bands cut from the gel (Figure 5) were deposited on carbon tape and dried. The samples were then placed in a vacuum experimental chamber and the location of the beam spot on a sample was determined using an optical microscope. A 2 MeV proton beam with a diameter of about 30 µm was applied and all samples were analyzed with a typical beam current of 200–300 pA with a collection time of 5 min per sample. The employed X-ray Amptek SDD detector (Si crystal 500 µm thick) was positioned at a distance of 3 cm from the sample. Its active area was 70 mm^2^, internally collimated to 50 mm^2^, and the input beryllium window (12.5 µm thick) enabled quantitative elemental analysis for elements from sodium (Na) upwards with an energy resolution of 125–135 eV. However, the low energy X–rays were suppressed by a Kapton filter (280 µm thick), therefore only elements of interest (roughly speaking, heavier than potassium (K)) were analyzed.

The X-ray spectra were acquired using the proprietary software and analyzed using the GUPIX code [67], which allowed quantification of the results. For this study, the most important information was to reliably confirm the presence of a detected element. Therefore, a positive detection was approved only if the calculated element concentration value exceeded the theoretical minimal detection limit (MDL) plus three single standard deviation values.

### 4.10. Atomic Emission Spectroscopy (AES)

The samples of a gel (Figure 5) were subjected to wet mineralization in a closed system using microwave energy. An Anton Paar Multiwave 3000 microwave system was used for mineralization. The samples were mineralized in a mixture of nitric acid (V) solution and H_2_O_2_ in a volume ratio of 1:3. The concentration of iron (Fe) and manganese (Mn) in the solutions obtained was determined by the atomic emission spectroscopy method using a Perkin Elmer Optima 7600 instrument. The wavelengths used to determine the concentration of the elements analyzed and the detection limits in relation to the methods used are listed in Table 2. To control the accuracy of the element analyses, a certified reference material, IAEA-V-10, was used. Table 2 shows the results of the analyses of the reference material and the recovery value based on the analyses performed in 3 replicates.

### 4.11. Statistical Analysis

The results presented in the study represent the means of three independent biological repetitions. The comparison of the significance of differences in mean values between the activity of SOD was performed using Tukey’s HSD test procedure at the level of significance *p* ≤ 0.01. The statistical analysis was performed using GraphPad v. 7.01 software (San Diego, CA, USA).

## 5. Conclusions

The obtained results of SOD activity staining, supported by spectroscopic analyses, provided evidence that the only active class of SODs in *Synechocystis* sp. PCC 6803, *G. violaceus* CCALA 979, and *Geitlerinema* sp. ZHR1A is FeSOD. In our opinion, the possible occurrence of Cu/ZnSOD in cyanobacteria, such as *Gloeobacter*, requires further studies.

## Figures and Tables

**Figure 1 ijms-23-13819-f001:**
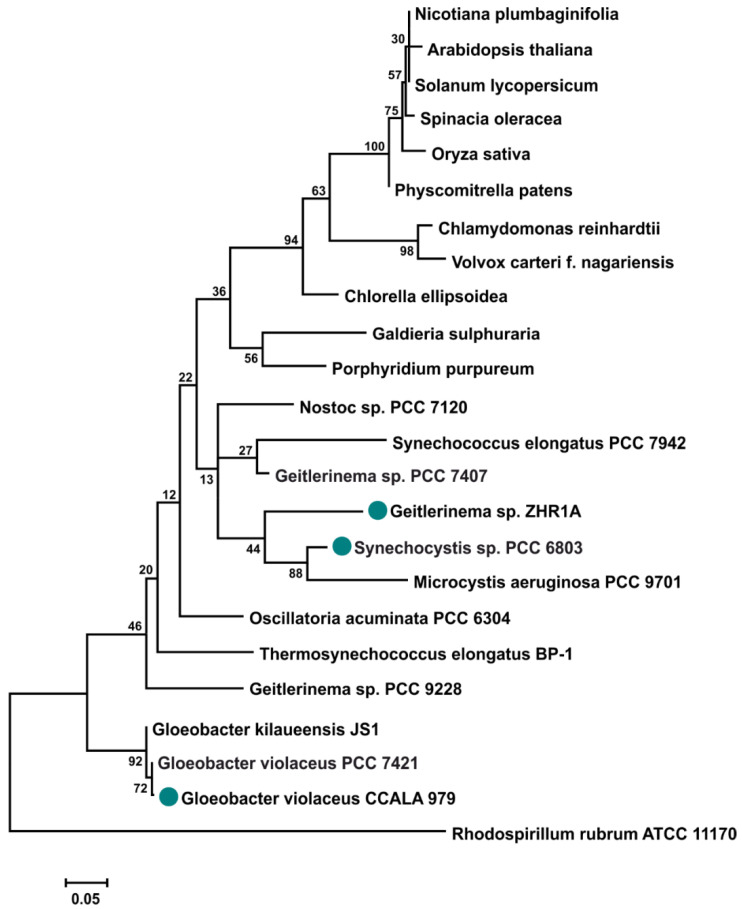
The 16S rRNA tree reconstructed for selected phototrophs performing oxygenic photosynthesis. The tree is rooted with 16S rRNA from *Rhodospirillum rubrum*. Blue-green circles denote cyanobacteria analyzed here, i.e., *Synechocystis* sp. PCC 6803, *Gloeobacter violaceus* CCALA 979 and *Geitlerinema* sp. ZHR1A (for details see Materials and Methods and Appendix A). Numbers at nodes show bootstrap values (based on 500 replicates). The scale represents the expected number of substitutions per nucleotide position.

**Figure 2 ijms-23-13819-f002:**
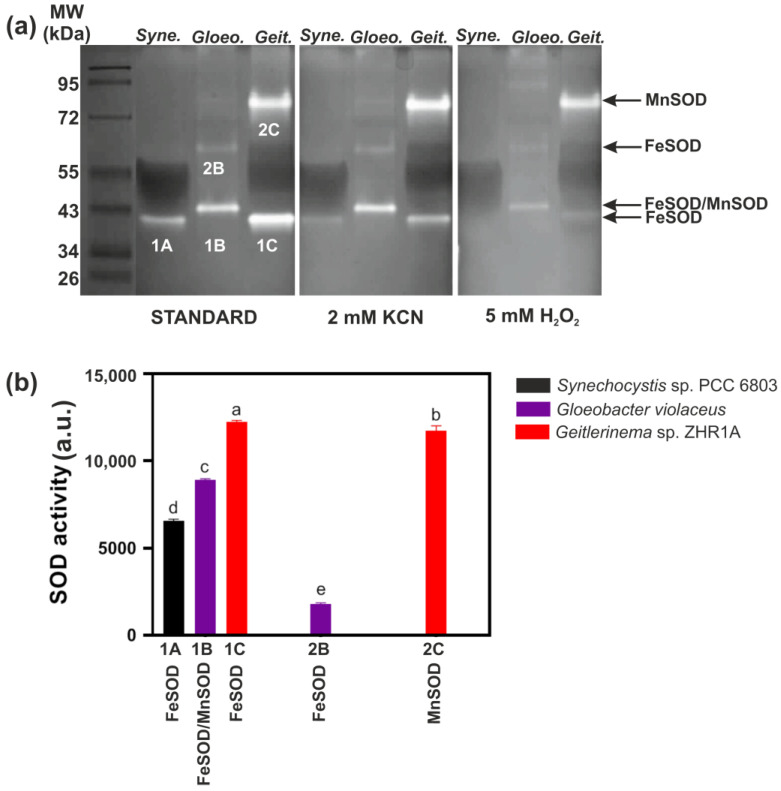
Activity of SOD forms in three species of cyanobacteria: *Synechocystis* sp. PCC 6803 (*Syne.*), *Gloeobacter violaceus* (*Gloeo.*), and *Geitlerinema* sp. ZHR1A (*Geit.*). (**a**) Identification of the different SOD forms in the assay with inhibitors: KCN and H_2_O_2_; “standard” denotes a standard staining buffer without inhibitors (see also “Materials and Methods”). 30 µg of protein was loaded on each lane. The in-gel activity bands of SOD from a standard staining buffer: 1A, 1B, 1C, 2B and 2C cut for spectroscopic analysis are shown. The lane with a protein-mass ladder is shown on the left. (**b**) Activity of the SOD forms without inhibitors expressed in arbitrary units (a.u.). Data represent means ± SD (*n* = 3). Statistical significance between means at least at *p* < 0.01 is indicated by different letters above the bars and was calculated by Tukey’s HSD test.

**Figure 3 ijms-23-13819-f003:**
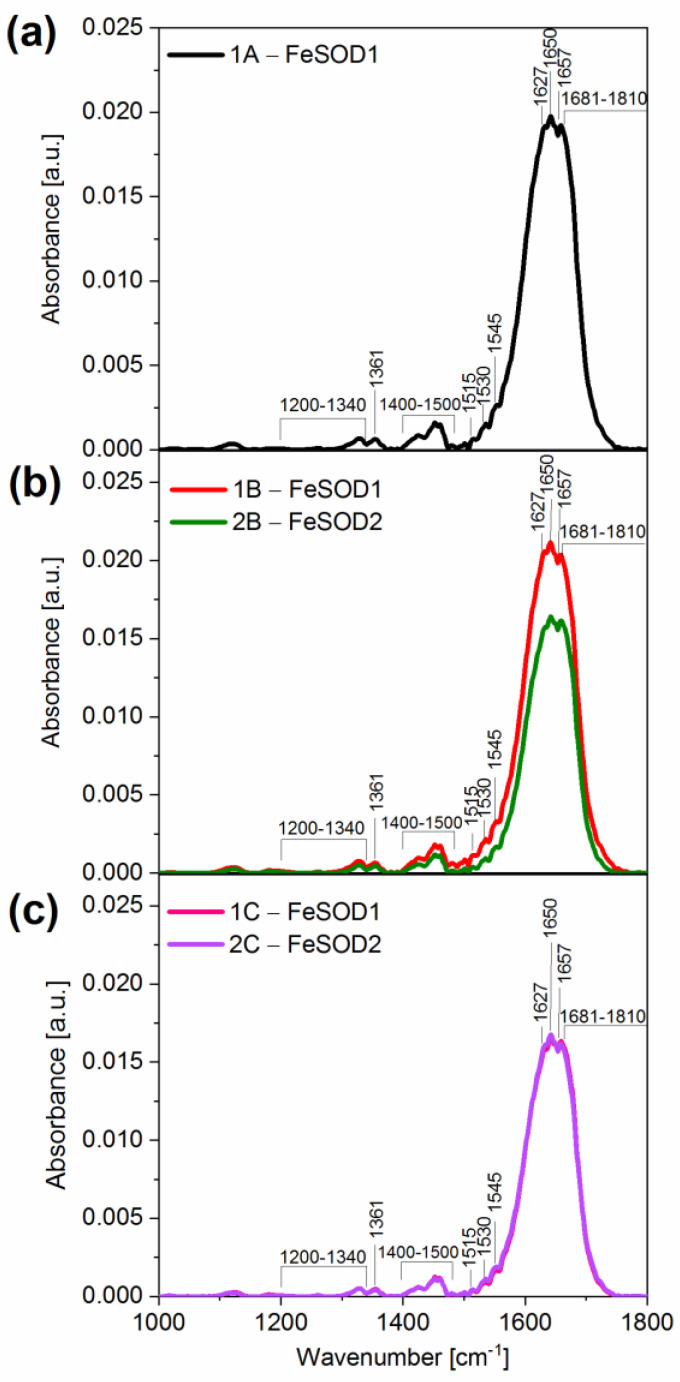
Fourier transformed infrared (FTIR) spectroscopy spectra of the bands of SOD, excised from the gel (see “Materials and Methods” and Figure 2). (**a**) Band 1A of *Synechocystis* sp. PCC 6803, (**b**) Bands 1B and 2B of *Gloeobacter violaceus*, and (**c**) Bands 1C and 2C of *Geitlerinema* sp. ZHR1A. These spectra are representative of three independent measurements.

**Figure 4 ijms-23-13819-f004:**
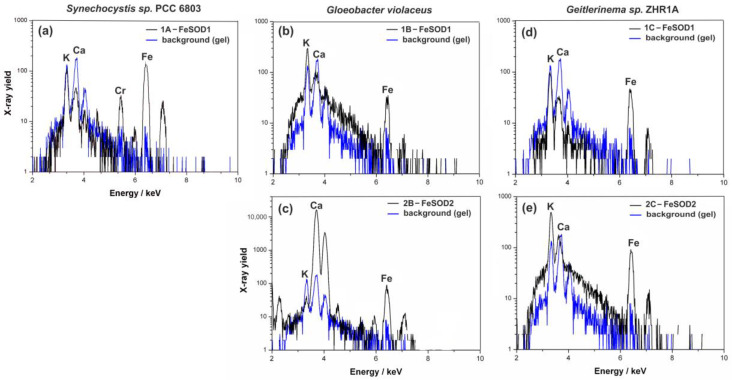
Particle-induced X-ray emission (PIXE) spectra with identified Kα peaks of the bands of SOD, excised from the gel (see “Materials and Methods” and Figure 2). (**a**) Band 1A of *Synechocystis* sp. PCC 6803, (**b**,**c**) Bands 1B and 2B of *Gloeobacter violaceus*, and (**d**,**e**) Bands 1C and 2C of *Geitlerinema* sp. ZHR1A. These spectra are representative of three independent measurements.

**Figure 5 ijms-23-13819-f005:**
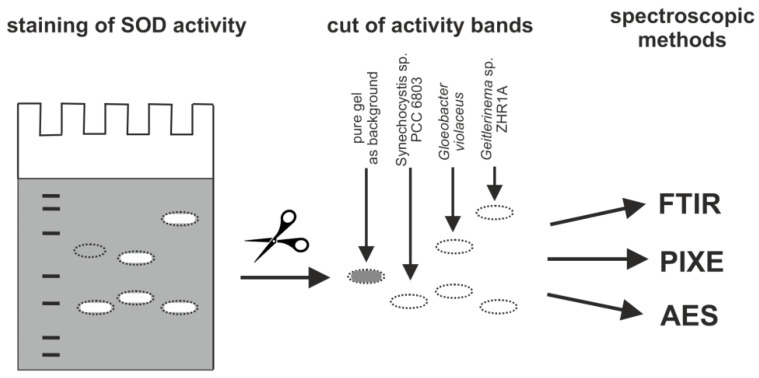
The scheme of preparation of SOD activity bands in a gel for spectroscopic analysis: Fourier transformed infrared (FTIR) spectroscopy, particle-induced X-ray emission (PIXE), atomic emission spectroscopy (AES). For the spectroscopic analyses, a pure gel without SOD activity was used as background.

**Table 1 ijms-23-13819-t001:** Content of iron (Fe) and manganese (Mn) in the bands of SOD of the analyzed cyanobacteria using the atomic emission spectroscopy (AES) method. The data represent means ± SD (*n* = 3).

Band of SOD	Strain	Fe (µg g^−1^)	Mn (µg g^−1^)	Identified Form of SOD
1A	*Synechocystis* sp. PCC 6803	5.0 ± 1.0	LOQ ^a^	FeSOD1
1B	*G. violaceus* CCALA 979	7.9 ± 2.7	LOQ	FeSOD1
2B		1.0 ± 0.4	LOQ	FeSOD2
1C	*Geitlerinema* sp. ZHR1A	9.6 ± 1.8	LOQ	FeSOD1
2C		2.6 ± 1.0	LOQ	FeSOD2

^a^ Limit of quantification (LOQ).

**Table 2 ijms-23-13819-t002:** Parameters of the applied atomic emission spectroscopy (AES) method.

Element	Wavelengths (nm)	Detection Limit (mg dm^−3^)	Content in Certificated Material (mg kg^−1^)	Measured (mg kg^−1^)	Recovery (%)
Fe	238.2	0.005	185	201.2	108.8
Mn	257.6	0.001	47	48.6	103.3

## Data Availability

Data sharing not applicable.

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
