# Peer review of "Application of Spectroscopic Methods for the Identification of Superoxide Dismutases in Cyanobacteria"

_ijms, 2022, doi:10.3390/ijms232213819_

Round 1

Reviewer 1 Report

This is a well written manuscript concerning the identification of the form of SOD protein expressed (Cu/Zn, Fe or MnSOD) in cyanobacteria using the typical gel-based activity assay coupled with spectroscopic methods.  The authors introduce the topic well, describe the evolution of organisms capable of oxygenic photosynthesis on Earth and so justify their experiments.  I really enjoyed this paper.

A couple of edits:

Lines 37-38 there are grammatical errors that make the meaning of this sentence difficult to understand.

Define FTIR, PIXE and AES first usage in the Results.

Table 1 please put the method used in the table title.

Were metals supplemented in the growth media?  Please state the media conditions in the materials and methods.

Author Response

Reviewer I

We would like to thank the reviewers for their valuable comments and constructive criticism of our manuscript. We have attached the detailed response below. All changes in the text of the manuscript are marked in red.

                                                                                                      Best regards

                                                                                                       Ireneusz Åšlesak

                                                                                                       (on-behalf of all co-authors)

Responses to the reviewer's comments (Reviewer I)

This is a well written manuscript concerning the identification of the form of SOD protein expressed (Cu/Zn, Fe or MnSOD) in cyanobacteria using the typical gel-based activity assay coupled with spectroscopic methods.  The authors introduce the topic well, describe the evolution of organisms capable of oxygenic photosynthesis on Earth and so justify their experiments.  I really enjoyed this paper.

A couple of edits:

Lines 37-38 there are grammatical errors that make the meaning of this sentence difficult to understand.

We have modified the sentence:

“O2·-  itself is not very toxic to cellular organic molecules, but it can be a source of other, more toxic ROS, especially hydroxyl radicals (HO·), which are very harmful to most biomolecules [1,2].”

Define FTIR, PIXE and AES first usage in the Results.

We have developed the abbreviations.

Table 1 please put the method used in the table title.

We have added this information to Table 1:

“Content of iron (Fe) and manganese (Mn) in the bands of SOD of the analysed cyanobacteria using the atomic emission spectroscopy (AES) method.”

 Were metals supplemented in the growth media?  Please state the media conditions in the materials and methods.

Cyanobacterial cultures were grown on a complete BG11 medium containing all micro- and macronutrients. Information on the medium used and the literature cited can be found on page 9.

Reviewer 2 Report

This manuscript is based on research of transition metals Fe/Mn like superoxide dismutases (SODS) in cyanobacteria by the spectroscopic techniques. I recommend its publication after minor revisions that are subjected to the editor decision.  

1-      Figure 3: The absorbance peaks at 1400 – 1500 cm-1 have not been identified yet, please comment about them.

2-      I suggest including a Conclusion of the Results that have been shown.  

Author Response

Reviewer II

We would like to thank the reviewers for their valuable comments and constructive criticism of our manuscript. We have attached the detailed response below. All changes in the text of the manuscript are marked in red.

                                                                                                      Best regards

                                                                                                     Ireneusz Åšlesak

                                                                                    (on-behalf of all co-authors)

Responses to the reviewer's comments (Reviewer II)

This manuscript is based on research of transition metals Fe/Mn like superoxide dismutases (SODS) in cyanobacteria by the spectroscopic techniques. I recommend its publication after minor revisions that are subjected to the editor decision.  

1-      Figure 3: The absorbance peaks at 1400 – 1500 cm-1 have not been identified yet, please comment about them.

Thank you very much for this comment. We did not mark the vibrations between 1400-1500 cm-1, because these are typical vibrations for the entire protein fraction, not only for SOD. But we think that all peaks that are visible should be marked. Therefore, in the new version of the figure, we added an area, and in the text, we added a fragment:

In each of the FTIR spectra shown in Fig. 3, the absorption bands at about 1400-1500 cm-1 are visible, which are attributed to asymmetric bending vibrations of the -CH2, -CH3 groups [35]. Moreover, the peaks at 1650 and 1545 cm-1 correspond to amide I (C = O stretching mode) and amide II (combination of N-H bending and C-N stretching mode) of SOD [36] and the peaks in the range 1681-1810 cm-1 and at the wavenumber 1361 cm-1 were attributed to the carbonyl (C=O) and methyl (C-H of -CH3) groups of synthetic iron-mineralized SOD (SOD-Fe0@Lapa-Z) [37].

2-      I suggest including a Conclusion of the Results that have been shown.  

We thank you for this comment. For more clarity, we have added the subsection “Conclusions” at the end of the ‘Discussion’.
